# Emergent representations in networks trained with the Forward-Forward algorithm

**Niccolò Tosato**[*]                                                                 *niccolo.tosato@phd.units.it*
*University of Trieste, Italy*
*AREA Science Park, Italy*

**Lorenzo Basile**[*]                                                                 *lorenzo.basile@phd.units.it*
*University of Trieste, Italy*
*AREA Science Park, Italy*

**Emanuele Ballarin**                                                             *emanuele.ballarin@phd.units.it*
*University of Trieste, Italy*

**Giuseppe de Alteriis**                                                         *giuseppe.de_alteriis@kcl.ac.uk*
*King's College London, UK*
*University College London, UK*

**Alberto Cazzaniga**                                                           *alberto.cazzaniga@areasciencepark.it*
*AREA Science Park, Italy*

**Alessio Ansuini**                                                             *alessio.ansuini@areasciencepark.it*
*AREA Science Park, Italy*

**Reviewed on OpenReview:** *https://openreview.net/forum?id=JhYbGiFn3Y*

## Abstract

The Backpropagation algorithm has often been criticised for its lack of biological realism. In an attempt to find a more biologically plausible alternative, the recently introduced *Forward-Forward* algorithm replaces the forward and backward passes of Backpropagation with two forward passes. In this work, we show that the internal representations obtained by the Forward-Forward algorithm can organise into category-specific *ensembles* exhibiting high sparsity – composed of a low number of active units. This situation is reminiscent of what has been observed in cortical sensory areas, where neuronal ensembles are suggested to serve as the functional building blocks for perception and action. Interestingly, while this sparse pattern does not typically arise in models trained with standard Backpropagation, it can emerge in networks trained with Backpropagation on the same objective proposed for the Forward-Forward algorithm.

## 1 Introduction

Deep Learning is a highly effective approach to artificial intelligence, with tremendous implications for science, technology, culture, and society. At its core, there is the Backpropagation (Backprop) algorithm (Rumelhart et al., 1986), which efficiently computes the gradients necessary to optimise the learnable parameters of an artificial neural network. Backprop, however, lacks biological plausibility (Stork, 1989) – leading to many attempts to address the issue. One of the most recent approaches, the Forward-Forward algorithm (Hinton, 2022), eliminates the need to store neural activities and propagate error derivatives along the network.

In a standard classification context, the application of Forward-Forward requires the designation of positive

---

[*]Equal contribution. Code is available at **https://github.com/NiccoloTosato/EmergentRepresentations**

and negative data. For example, to classify images, one could assign positive (or negative) data to those images having their correct (or incorrect, respectively) class label embedded via one-hot encoding at the border (as shown in Figure 1, Panel **A**). The Forward-Forward algorithm then learns to discriminate between positive and negative data by optimising a goodness function (*e.g.*, the $\ell_2$ norm of the activations), akin to contrastive learning (Chen et al., 2020). Satisfactory results have been observed (Hinton, 2022) for classification tasks on MNIST (Lecun et al., 1998), a standard benchmark dataset. This work takes a step beyond performance evaluation, delving into the structure of the hidden representations learned by the Forward-Forward algorithm, uncovering their spontaneously sparse nature and drawing parallels to neural *ensembles* observed in the brain (Miller et al., 2014; Yuste et al., 2024).

We organise this paper as follows. In section 2 we set the stage by providing a brief overview of the Forward-Forward algorithm and of neuronal ensembles. Then, section 3 is dedicated to the description of the models and datasets investigated and the methods used to analyse representations. Our analysis of the Forward-Forward representations begins in section 4, where we present our key findings. Specifically, in subsection 4.2, we show that the Forward-Forward algorithm spontaneously learns sparse representations, organised into artificial ensembles, *i.e.,* small sets of highly specialised neurons that consistently co-activate for data in a given class. In subsection 4.3, we demonstrate that these ensembles can overlap, with individual units contributing to multiple ensembles when visual features are shared. Further, subsection 4.4 reveals that ensembles can arise on previously unseen categories, indicating a robust generalization of this representational mechanism. Notably, these ensembles can share units with those associated with seen categories, demonstrating effective integration of new information with concepts learned during training. Finally, in subsection 4.5, we examine the structure of the weights and show that the observed sparsity and ensemble formation arise from suppression mechanisms, analogous to the inhibitory processes mediated by biological neurons (Yuste, 2015). These findings are particularly striking because the Forward-Forward algorithm achieves these properties without requiring explicit regularisation to induce sparsity. We observe that, although optimising the cross-entropy loss for the same classification task does not appear to produce the sparse ensembles we observe, the phenomenon may not solely be due to the use of the Forward-Forward algorithm. In fact, similar results are obtained by optimising the same goodness function of Forward-Forward, with Backprop instead. This suggests that more focus should be put on the purpose and biological meaning of the loss function rather than the training algorithm (Richards et al., 2019). We discuss our results in section 5.

In summary, our main results are as follows:

- The Forward-Forward algorithm yields sparse representations composed of small groups of highly specific units, which we refer to as ensembles by analogy with those observed in the cortex.

- Ensembles can emerge in a zero-shot manner for classes held out during training and they can share units across visually related categories.

- The emergence of sparsity and the formation of ensembles are not unique to Forward-Forward optimisation, as they can also be observed in networks trained with Backpropagation on the same objective function.

## 2 Related Work

In the section that follows, we summarise key aspects of the Forward-Forward algorithm and the main findings pertaining identification and characterisation of biological neuronal ensembles in the brain.

### 2.1 Forward-Forward

The Forward-Forward algorithm (Hinton, 2022) is a recently proposed learning algorithm for artificial neural networks, whose main premise is the ability to overcome the notorious biological implausibility of Backprop (Rumelhart et al., 1986). In fact, while the effectiveness of Backprop makes it the standard algorithm for training neural networks, it is based on biologically unrealistic assumptions, such as the need to propagate information forwards and backwards through the network (Richards et al., 2019).

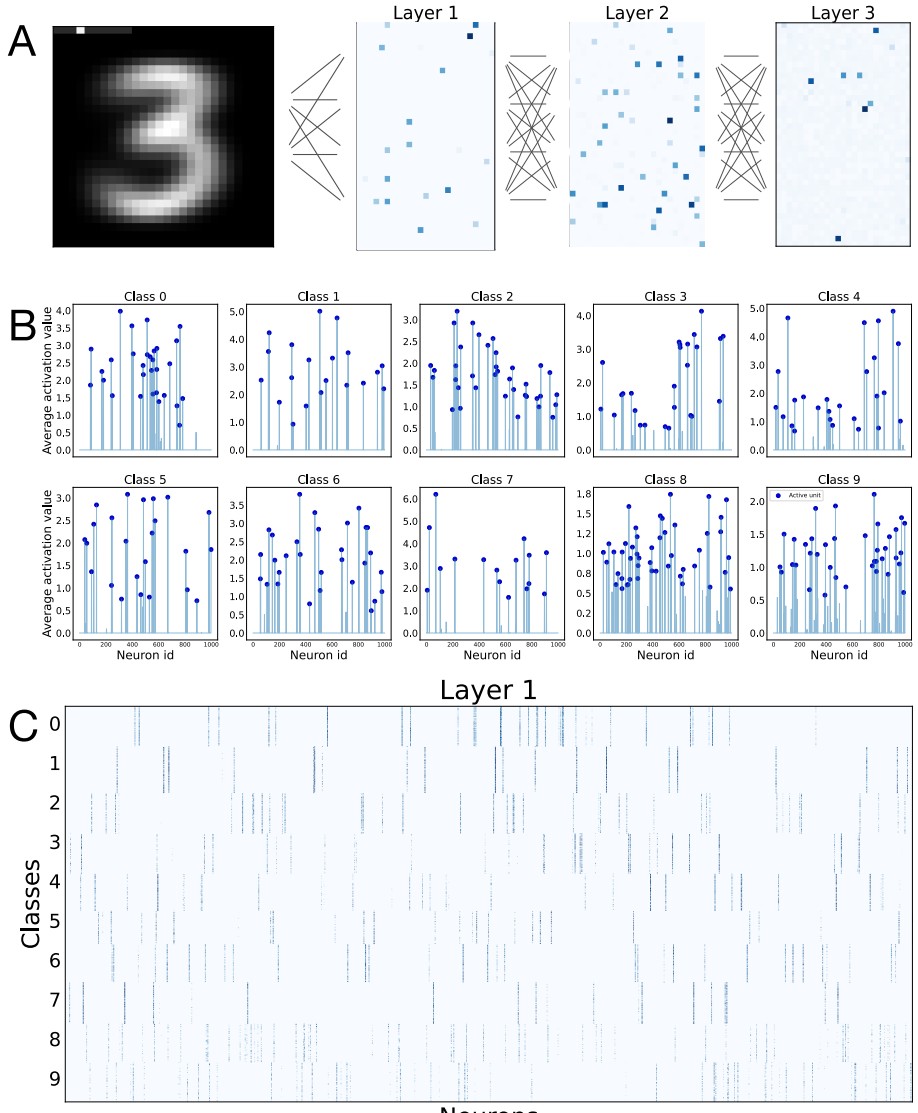

Figure 1: Activation patterns in a Multi-Layer Perceptron trained with the Forward-Forward algorithm, on the Mnist dataset.

Panel **A** Examples of activation patterns in response to a positive input (class label embedded as a one-hot encoding on the top left corner of the image). Images show the activation value for network units, arranged as a matrix only for the sake of clarity; darker squares represent more active neurons.

Panel **B** Activation value of each neuron in the first hidden layer (Layer 1), averaged on all images of a given class. Neuron index on the $x$ axis; average activation on the $y$ axis. Blue dots indicate units that are considered active according to the leave-one-out (LOO) method described in subsection 3.5.

Panel **C** Activation map for neurons in Layer 1 for all images, grouped by class. A blue dot in position $(x, y)$ indicates that neuron $x$ is activated by input $y$; colour scale represents the intensity of such activation. Horizontal bands mark different categories; blue vertical *stripes* mark active, category-specific neurons. Each input category activates consistently a specific sets of neurons (ensemble).

Forward-Forward owes its name to the fact that it replaces the backward pass with an additional forward pass. The two forward passes are executed on different data, named positive and negative data. During training, the objective of Forward-Forward is to maximise a so-called goodness function of the neural activations (*e.g.,*

the $\ell_p$ norm) on positive data and minimise it on negative data. In a simple image classification setting, such as the one we adopt in this paper, one could encode a class label at the border of images, by one-hot encoding it with a white pixel (as shown in Figure 1, Panel **A**). Then, following the definition from Hinton (2022), positive data are those for which the encoded label matches the ground truth label, while the opposite holds for negative data. Layers are trained separately and sequentially, and learn to discriminate between positive and negative data by maximising and minimising their goodness, according to the data presented. Crucially, activations are normalised before being passed to the subsequent layer, to prevent layers from relying on the goodness computed by their predecessors. At test time, when a new unlabelled sample has to be categorised, many copies of the image are created, each with a different one-hot encoded label. These are then fed into the neural network to obtain a goodness score. Finally, the image gets classified in the category that produced the maximum goodness value.

In the seminal Forward-Forward paper (Hinton, 2022), satisfactory classification results are reported on the standard handwritten digit recognition dataset MNIST, with the definition of positive and negative data described above, and using the $\ell_2$ or $\ell_1$ norm of activations as goodness function. In a recent theoretical work, it has been analytically shown that under somewhat mild assumptions sparsity emerges in Forward-Forward layers (Yang, 2023) as a consequence of optimising the Forward-Forward loss. While these formal results are derived for a single layer, they offer a theoretical grounding for our experimental findings. Recent studies inspired by the Forward-Forward learning procedure have expanded its applicability across various architectures, notably achieving enhanced performance in Convolutional Neural Networks (CNNs) (Papachristodoulou et al., 2024; Sun et al., 2025). Additionally, other works have proposed alternative goodness functions and explored the specific contributions of individual neurons to the classification process, shedding light on the interpretability and adaptability of the approach (Terres-Escudero et al., 2024).

We illustrate properties of representations obtained in Forward-Forward networks, that are reminiscent of what is found the neocortex and hippocampus, where ensembles of a few number of units activate consistently in response to similar stimuli. We discuss properties of neuronal ensembles in the following section.

## 2.2   Neuronal ensembles

In Neuroscience, neuronal ensembles are defined as sparse groups of neurons that co-activate either spontaneously or in response to sensory stimuli. These ensembles, rather than individual neurons, have long been proposed as emergent functional units of cortical activity, playing critical roles in sensory processing, memory, and behaviour (Miller et al., 2014; Hebb, 2005; Harris, 2005; György, 2010; Hopfield, 1982; Carrillo-Reid et al., 2019; Carrillo-Reid & Yuste, 2020; Yuste et al., 2024). Recent reviews, such as Yuste et al. (2024), provide a comprehensive overview of the concept and its implications.

The importance of ensembles has been increasingly corroborated by experimental studies, enabled by advances in techniques like calcium imaging, which allow for simultaneous recording of large-scale neural activity at single-cell resolution (Carrillo-Reid & Calderon, 2022). For example, Miller et al. (2014) demonstrated that, during visual processing, cortical spiking activity is dominated by ensembles whose properties cannot be explained by the independent activity of individual neurons. These ensembles are activated both by sensory stimuli (*e.g.,* visual inputs) and by spontaneous network activity, suggesting that they represent intrinsic functional building blocks of cortical responses. Notably, single neurons often participate in multiple ensembles, thereby enhancing the network's encoding potential (Rigotti et al., 2013; Fusi et al., 2016). Further evidence from Yoshida & Ohki (2020) showed that sparse ensembles in the primary visual cortex (V1) are elicited by visual stimuli. Images can be decoded reliably from the activity of a small subset of highly responsive neurons, with additional neurons either failing to improve or even degrading decoding performance. These findings underscore the efficiency of sparse representations, likely facilitated by partially overlapping receptive fields. This arrangement enables robust and efficient encoding of visual information, making sparse ensembles an optimal strategy for downstream processing.

The presence and functionality of ensembles are not limited to specific species or sensory modalities. Studies in various animal models have revealed their role in diverse neural processes (Dupre & Yuste, 2017; Liu & Baraban, 2019), and recent findings suggest they may even contribute to conscious experience (Boyce et al., 2023). Moreover, ensembles have demonstrated remarkable stability over time. For instance, Pérez-Ortega

et al. (2021) showed that neuronal ensembles can persist for weeks, supporting their potential involvement in long-term representations of perceptual states or memories.

Technological advancements have also enabled not just the visualization but the direct stimulation of ensembles, allowing to "play the piano" with ensembles of neurons Carrillo-Reid & Yuste (2020). All-optical approaches, such as those described by Packer et al. (2015) and Carrillo-Reid et al. (2019), have shown that repeatedly stimulating specific groups of neurons in V1 can imprint ensembles that remain spontaneously active even after a day. These imprinted ensembles exhibit pattern completion, where activating a subset of neurons can recall the entire ensemble. Remarkably, this effect persists long after the initial stimulation, and experiments have demonstrated causal links between ensemble activation and behaviour (Carrillo-Reid & Yuste, 2020).

Finally, the concept of neuronal ensembles has inspired computational models. For instance, Doi & Lewicki (2004) demonstrated that sparse and redundant representations are optimal for encoding natural images, particularly when neurons are unreliable, a result corroborated by earlier studies (Field, 1994; Olshausen & Field, 2004). These computational frameworks align with biological observations, suggesting that sparse ensemble representations are both efficient and robust mechanisms for encoding sensory information.

## 3 Methods

In this work, we investigate and compare the representations produced by three models [1]:

- A classifier in the style of that used by Hinton (2022), trained with Forward-Forward (**FF**);
- A classifier identical to the above, but trained end-to-end with Backprop to optimise the same goodness function (**BP/FF**);
- A classifier trained with Backprop on the categorical cross-entropy loss, as customary (**BP**).

Such different scenarios are described individually in subsection 3.2, subsection 3.3 and subsection 3.4, respectively.

### 3.1 Data

The datasets we use to train and test the models described so far are MNIST (Lecun et al., 1998), FASHIONMNIST (Xiao et al., 2017), SVHN (Netzer et al., 2011) and CIFAR10 (Alex, 2009). Details on these datasets are provided in the Appendix B.

### 3.2 Model trained with Forward-Forward (FF)

Our **FF** model is inspired by the architecture proposed by Hinton (2022) – and likewise trained according to the Forward-Forward algorithm. It consists of three fully-connected layers, each composed by 1000 units in the case of MNIST and FASHIONMNIST, and 3072 units in the case of SVHN and CIFAR10. Each linear layer is followed by elementwise ReLU non-linearities. Both during training and inference, the layer-wise $\ell_2$ norm is used as the goodness function of choice; correspondingly, $\ell_2$ normalisation is performed between subsequent layers. Additional results obtained using the $\ell_1$ norm as a goodness function are presented in Appendix K.
To define positive and negative data, a one-hot-encoded class vector is embedded at the top-left corner of images. Prior to such embedding, these pixels are set to black colour. Then, in the case of positive data, the pixel corresponding to the true class is switched to the maximum value elsewhere observed in the image, while in the case of negative examples such value is randomly assigned to one of the other pixels of the embedding vector.
During training, the weights are optimised by minimising the loss function $L = \log(1 + e^{G_{neg} - G_{pos}})$, where $G_{neg}$ and $G_{pos}$ are, respectively, the goodness value for negative and positive data. At inference time, for

---

[1]From this point on, we use the term *model* to refer to the combination of network architecture and optimisation algorithm.

each layer, the goodness values corresponding to every possible label are converted into a probability using softmax. By performing this step for each layer, they can contribute equally to the prediction.

For comprehensive details on the training procedures of the models discussed in this section and the next two sections, we direct the reader to Appendix C.

### 3.3   Model trained with Backpropagation on the goodness objective (BP/FF)

The architecture of the **FF** model, while designed to be optimised using the Forward-Forward algorithm, can be trained seamlessly with Backprop on the same goodness maximisation/minimisation objective. Indeed, keeping the definition of positive and negative data introduced for **FF**, one could simply use Backprop to optimise the goodness-based loss from the Forward-Forward algorithm.
In detail, positive and negative data are fed to the network during the forward step, and the overall goodness of the internal representation is evaluated. The backward pass is then executed, and parameters are optimised to achieve the same goal as the **FF** model. It is worth pointing out that, in this case, the goodness is maximised globally instead of layer-by-layer (*i.e.*, locally).

### 3.4   Model trained with Backpropagation on the cross-entropy loss (BP)

The **FF** and **BP/FF** models are also compared to a standard neural classifier, serving as a baseline. For such purpose, a multi-layer perceptron is employed. The model shares the same number of layers, layerwise neuron count, and non-linear activation function choice with **FF** and **BP/FF**. The only architectural difference between the **BP** model and the other two is the addition of a final softmax layer, to suitably shape and scale the output for the classification task. The model is trained end-to-end with Backprop on the categorical cross-entropy loss.

### 3.5   Analysis of representations

For each model described, we analyse the internal representation emerging at each layer. We limit our analysis to data belonging to the test set (*i.e.,* not seen during training) and correctly classified by the respective model. However, the main results of our analysis, concerning sparse and ensemble-like representations, extend without any modification to training data. Concretely, the representation of a single image is a $n$-dimensional vector composed by the activations (after the ReLU non-linearity) of all the units in the layer. For each layer, we extract a representation matrix $X$ of size $(M, n)$, where $M$ is the total number of test images (correctly classified) and $n$ is the number of neurons in the layer considered.

**Sparsity**

For each representation vector $x$ we assign a *sparsity* measure following the notion of sparsity introduced in Hoyer (2004):

$$S(x) = \frac{\sqrt{n} - \frac{\|x\|_1}{\|x\|_2}}{\sqrt{n} - 1}$$

With this definition, when $S(x) = 1$ the vector $x$ contains only one non-zero component representing the case of an extreme sparsity. The other limiting case is the one in which all the components of $x$ are equal in magnitude, in this case $S(x) = 0$. The sparsity function $S$ interpolates smoothly between these two extremes. The sparsity of a layer representation is obtained by averaging the sparsity of its component vectors $S = \frac{1}{M} \sum_{i=1}^{M} S(x_i)$.

**Ensembles**

To detect the emergence of category-specific ensembles, within each model and dataset combination, we adopt the following method. The idea is that a neuron should be considered active and part of an ensemble if it activates consistently and specifically when the network receives input data that belongs to that category. We start by defining a category-specific representation matrix $X_c$, of shape $(M_c, n)$, where $M_c$ is the number of

correctly classified test images of the given category. Then, we compute the average activation of each hidden unit across all samples: $\overline{x_{j,c}} = \frac{1}{M_c} \sum_{i=1}^{M_c} (X_c)_{ij}$; and the leave-one-out average of the averages $\text{LOO}_{j,c} = \frac{1}{n-1} \sum_{i \neq j} \overline{x_{j,c}}$. We then classify a neuron $i$ as active (*i.e.,* part of an ensemble) if $\overline{x_{i,c}} > 2 \cdot \text{LOO}_{i,c}$. We also perform a significance test for these comparisons through a permutation test (see Appendix D and Table 3). Examples of average activation profiles and of ensembles are reported in Figure 1.

The output of the ensemble computation is a set of active units for each category: $\mathcal{E}^c = \{e_1^c, e_2^c, \dots e_{n_c}^c\}, \forall c \in \{1, 2, \dots, C\}$, where $n_c$ is the number of active units for category $c$. Once the ensembles are defined, it is possible to look at units that are shared across categories $c$ and $c'$ by considering $\mathcal{E}^c \cap \mathcal{E}^{c'}$. The size of the shared units is naturally measured by $| \mathcal{E}^c \cap \mathcal{E}^{c'} |$.

We can also measure the similarity between two ensembles using the Jaccard similarity index (intersection over union): $J(\mathcal{E}^c, \mathcal{E}^{c'}) = \frac{|\mathcal{E}^c \cap \mathcal{E}^{c'}|}{|\mathcal{E}^c \cup \mathcal{E}^{c'}|}$. As an example, for two ensembles composed of 50 units with a substantial ensemble overlap of 30% (15 units), the similarity $J$ is $\approx 0.18$. Examples of shared units are reported in Figure 3 (see Table 3 for typical ensemble sizes in our setting).

When the sparsity $S$ of a representation is low, ensembles are typically ill-defined as too many neurons are significantly active simultaneously and the notion of active unit tends to blur. To set a threshold, we will consider values of $S$ below 0.5 as non-sparse, and in these cases, we do not define ensembles out of the representation.

# 4 Results

In this section, we describe our findings for the three models introduced, on the MNIST, FASHIONMNIST, SVHN and CIFAR10 datasets. In particular, we focus on the properties of representations obtained within the **FF** model, *i.e.,* a model trained with Forward-Forward on its natural goodness objective. Such properties, such as the emergence of category-specific ensembles and the presence of shared units across them, establish a link between neural networks trained with the Forward-Forward algorithm and biological cortical networks described in subsection 2.2.

## 4.1 Classification accuracy

Before we present the main results of this work, we evaluate the performances of our models on the classification tasks at hand. Table 1 contains results in terms of test set classification accuracy for all models we employed – **FF**, **BP/FF** and **BP** – on MNIST, FASHIONMNIST, SVHN and CIFAR10. While some of these accuracy values are far from the state-of-the-art (*i.e.,* respectively, 0.997 (Cireşan et al., 2010), 0.931 (Xiao et al., 2017), 0.860 (Pitsios, 2017) and approximately 0.7 (Lin et al., 2015), for fully-connected networks), they are a solid ground on which to build our subsequent investigations. Training details and hyperparameters for all models are reported in Appendix C.

Table 1: Test-set classification accuracy for the models considered in our investigation. Results expressed as *mean $\pm$ std. dev.* over 10 runs with independent randomised weight initialisation.

| Dataset | **FF** | **BP/FF** | **BP** |
|---|---|---|---|
| MNIST | $0.94 \pm 0.008$ | $0.969 \pm 0.001$ | $0.982 \pm 0.001$ |
| FASHIONMNIST | $0.849 \pm 0.002$ | $0.877 \pm 0.002$ | $0.892 \pm 0.004$ |
| SVHN | $0.716 \pm 0.002$ | $0.799 \pm 0.004$ | $0.793 \pm 0.145$ |
| CIFAR10 | $0.484 \pm 0.004$ | $0.521 \pm 0.006$ | $0.564 \pm 0.004$ |

## 4.2 Forward-Forward elicits sparse neuronal ensembles

The **FF** and **BP/FF** models – based on the original Forward-Forward network architecture, and trained according to the goodness objective (subsection 3.2 and subsection 3.3) – exhibit typically high sparsity levels in their representations, in clear contrast with **BP** (see Figure 2 and Table 2). While sparsity does not

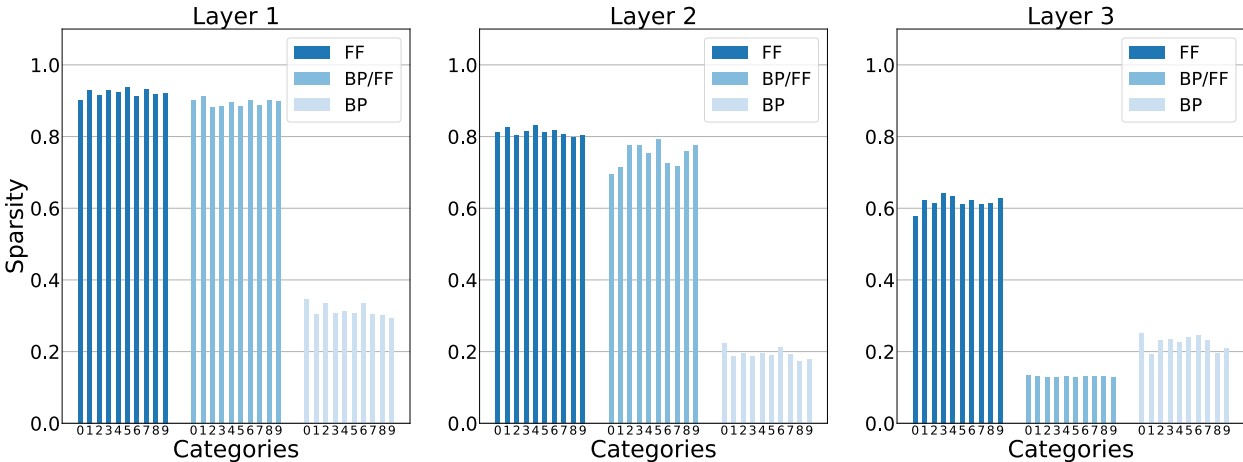

Figure 2: Sparsity of category-specific representations. We report the sparsity of representations - computed as described in subsection 3.5 - for the three models **FF**, **BP/FF** and **BP** on the MNIST dataset. Sparsity values are the average over 10 runs.

spontaneously arise in **BP**, it can be enforced by means of $\ell_1$ regularisation of the activations (Georgiadis, 2019). An analysis of this setting is presented in Appendix I.

Table 2: Average sparsity for all combinations of model, dataset and layer, according to the definition given in subsection 3.5. Results are expressed as *mean ± std. dev.* computed over 10 runs with independent random weights initialisation.

| Model | Layer | MNIST | FASHIONMNIST | SVHN | CIFAR10 |
|-------|-------|-------|--------------|------|---------|
| **FF** | 1 | $0.922 \pm 0.001$ | $0.85 \pm 0.002$ | $0.83 \pm 0.001$ | $0.77 \pm 0.001$ |
| | 2 | $0.813 \pm 0.019$ | $0.605 \pm 0.015$ | $0.706 \pm 0.001$ | $0.728 \pm 0.002$ |
| | 3 | $0.618 \pm 0.074$ | $0.628 \pm 0.013$ | $0.489 \pm 0.004$ | $0.566 \pm 0.002$ |
| **BP/FF** | 1 | $0.895 \pm 0.005$ | $0.81 \pm 0.007$ | $0.783 \pm 0.003$ | $0.753 \pm 0.004$ |
| | 2 | $0.747 \pm 0.013$ | $0.851 \pm 0.007$ | $0.95 \pm 0.003$ | $0.932 \pm 0.003$ |
| | 3 | $0.131 \pm 0.011$ | $0.065 \pm 0.009$ | $0.133 \pm 0.011$ | $0.135 \pm 0.009$ |
| **BP** | 1 | $0.315 \pm 0.003$ | $0.352 \pm 0.003$ | $0.47 \pm 0.02$ | $0.478 \pm 0.016$ |
| | 2 | $0.193 \pm 0.004$ | $0.241 \pm 0.005$ | $0.524 \pm 0.212$ | $0.3 \pm 0.18$ |
| | 3 | $0.225 \pm 0.006$ | $0.248 \pm 0.006$ | $0.232 \pm 0.106$ | $0.164 \pm 0.006$ |

When the sparsity level is sufficiently high ($S > 0.5$) we are able to identify small sets of neurons (ensembles) that consistently co-activate across all the samples of the same class, similar to what has been observed in cortical representations (Yuste, 2015; Miller et al., 2014; Harris, 2005).

Figure 1 (Panels **B**, **C**) shows an example of average neuron activations for each class in Layer 1 of the **FF** model trained on MNIST, and showcases the emergence of sparse, category-specific, ensembles (see the Appendix E for a similar visualisation for Layers 2 and 3 of the same model and Appendix F for Layer 1 in all the models). These representations typically activate only a small fraction of units: ensembles consisting of just a few percent of the neurons in a layer are commonly observed, whether working with simpler datasets (e.g., MNIST, FASHIONMNIST) or more complex ones (SVHN, CIFAR10), with a slight tendency of the **FF** model to create larger ensembles in the latter case (Table 3).

Overall, these findings show that networks trained with the Forward-Forward objective produce highly sparse representations, characterised by *ensembles*, *i.e.,* small groups of neurons with category-specific activation patterns.

Table 3: Average fraction of units taking part in ensembles, for all combinations of dataset and layers, in the **FF** and **BP/FF** models. Ensemble sizes are averaged across all categories, divided by the number of neurons in a layer, and then expressed in %. Ensembles are defined according to the LOO method presented in subsection 3.5. Results expressed as *mean $\pm$ std. dev.* . In the third layer of **BP/FF**, as well as in **BP**, the representation is non-sparse. With (*) we marked the condition in which $\approx 2.6\%$ of the units have a $p$-value larger than 0.05, see Appendix D for details.

| Model | Layer | Mnist | FashionMnist | Svhn | Cifar10 |
|---|---|---|---|---|---|
| **FF** | 1 | $3.69 \pm 0.09$ | $5.02 \pm 0.14$ | $10.3 \pm 0.15$ | $16.08 \pm 0.09$ |
| | 2 | $5.31 \pm 0.35$ | $18.46 \pm 0.66$ | $21.28 \pm 0.23$ | $21.2 \pm 0.3$ |
| | 3 | $1.36 \pm 0.36$ | $20.59 \pm 0.63$ | $4.48 \pm 0.52$ | $4.86 \pm 0.51$ |
| **BP/FF** | 1 | $8.58 \pm 0.23$ | $13.24 \pm 0.31$ | $15.07 \pm 0.16$ | $13.3 \pm 0.13$ |
| | 2 | $13.18 \pm 0.67$ | $8.45 \pm 0.47$ | $5.08 \pm 0.19$ | $5.55 \pm 0.28(*)$ |
| | 3 | - | - | - | - |

### 4.3 Visually similar classes can elicit ensembles with shared neurons

Drawing a parallel with a phenomenon observed in Neuroscience (Yoshida & Ohki, 2020), related categories can be expected to share units of their ensembles. This is indeed what we observe, as shown in Figure 3. Results are reported for FashionMnist, where different classes of clothes or shoes may contain a common share of visual features. In this regard, we observe a clear tendency to share units between similar classes – *e.g.,* across representations of `pullover`, `coat` and `shirt`.

In the following section, we provide evidence that a unit can be shared across two ensembles even if one of these refers to an unseen category (*i.e.,* excluded from the training set but whose representation, extracted at test time, generates an ensemble), as we show in Figure 4 (Panel **C**).

These findings indicate that ensembles relative to visually related classes can partially overlap.

### 4.4 Representations of unseen categories can elicit well-defined ensembles

We investigate the ability of a trained **FF** model to respond to unseen categories with a coherent activation pattern which is typical of the ensembles we found on the categories seen at training time. To this end, we repeatedly train **FF** on FashionMnist, removing one category at a time. Then, we extract the representation of the missing category, and verify if an ensemble is formed. We find that in all the ten cases, this is indeed the case, and the new ensemble share the same characteristics of the ones emerging for seen categories, apparently with the only exception of a lower average activation of their constituent units (see Figure 4 for one example, and the Appendix G for a more detailed account).

In several cases, we also find that the ensembles of unseen categories share units with the ensembles of seen categories, when endowed with similar visual features (Figure 4, Panel **C**). A more extensive exploration of these cases is also reported in the Appendix G.

These results show that ensemble-like structures can arise in a zero-shot setting on data categories held out of the training set.

### 4.5 Distribution of excitatory and inhibitory connections

As we observed in subsection 4.2, **FF** and **BP/FF** have comparable sparsity levels and, when they are defined, the ensembles have comparable sizes. A more fine-grained inspection of the representations learned by different models at different layers can be found in Appendix J. The presence of sparse ensembles suggests that a strong inhibition mechanism is at work, leaving only a few neurons active for each data sample. Inhibition in these architectures is the result of an interplay between the sign and magnitude of the weights and the those of the biases. Therefore it is natural to wonder whether **FF** and **BP/FF** are similar also in this interplay between weights and biases. We find that the answer is *no*: the **FF** and **BP/FF**

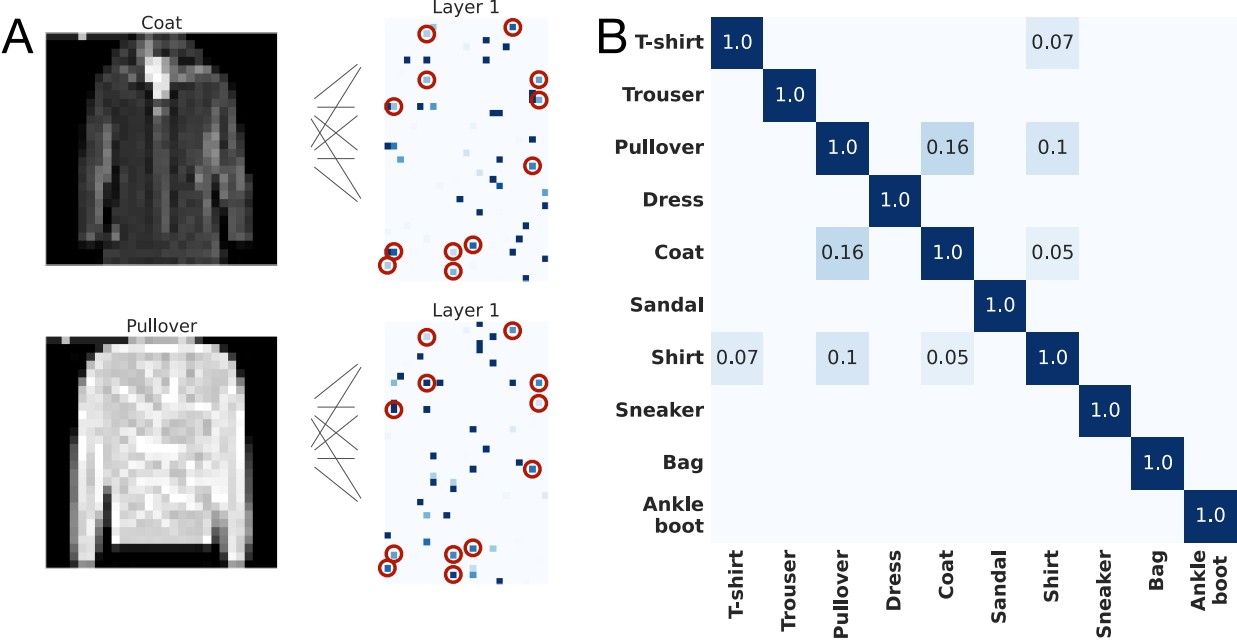

Figure 3: Visually similar classes in FASHIONMNIST can elicit ensembles with shared neurons.
Panel **A** The ensembles elicited in the first hidden layer of **FF** by two example inputs. Red circles indicate the active units which are shared between the two categories.
Panel **B** Element $i, j$ of the matrix indicates how many units are shared between the ensembles of category $i$ and category $j$ (normalised by the ensemble sizes), by using the Jaccard similarity index: $J(\mathcal{E}^i, \mathcal{E}^j) = \frac{|\mathcal{E}^i \cap \mathcal{E}^j|}{|\mathcal{E}^i \cup \mathcal{E}^j|}$. The results are referred to a single training run.

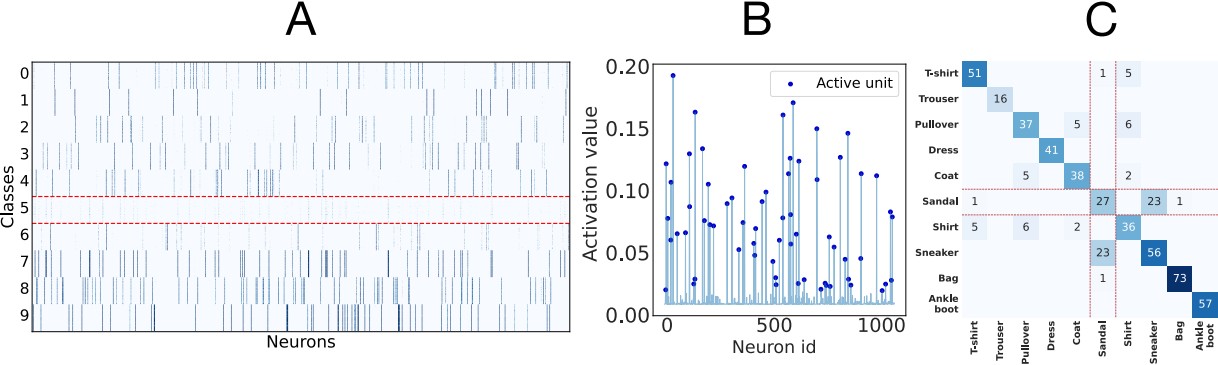

Figure 4: The representations of an unseen category form an ensemble in **FF** trained on FASHIONMNIST.
Panel **A** Activation patterns in response to the different categories in the first hidden layer. The unseen category (`Sandal`), surrounded by red lines, produces a relatively weaker but well-defined ensemble-like activation pattern.
Panel **B** Activation value of each neuron, averaged on all images of the unseen category. Neuron index on the $x$ axis; average activation on the $y$ axis. Blue dots indicate units that are considered active according to the method described in subsection 3.5.
Panel **C** Ensembles of unseen categories can share units with the ensembles of the other categories. Element $i, j$ of the matrix indicates how many units are shared between the ensembles of category $i$ and category $j$: $| \mathcal{E}^i \cap \mathcal{E}^j |$. The results are referred to a single training run.

are indeed two profoundly different models that create sparse representations and ensembles with different mechanisms.

To show this, we consider for each neuron $i$ in a layer with width $n$ the fraction of its positive weights *w.r.t.* the total number of its input connections ($\varrho_i^+$ in the following), and its bias $\beta_i$. From these neuron-level quantities we construct their layer averages: $\varrho^+ = \frac{1}{n} \sum_i \varrho_i^+$ and $\beta = \frac{1}{n} \sum_i \beta_i$. The neuron's weights are strongly imbalanced towards inhibition when $\varrho_i^+ \approx 0$ and, *viceversa*, strongly imbalanced towards excitation when $\varrho_i^+ \approx 1$; when $\varrho_i^+ \approx 0.5$ we will say that the neuron's weights are almost perfectly balanced; similar considerations hold for the biases, where a large and negative $\beta_i$ means strong inhibition for the $i$-th unit.

Focusing on the second hidden layer, we observe macroscopic differences in the empirical distribution of $\varrho_i^+$ among the three models (see Figure 5), with 1) a dominance of positive weights in the case of **FF**, 2) a bimodal distribution of $\varrho_i^+$ in **BP/FF**, with two populations of imbalanced units in opposite directions, and 3) a unimodal and approximately balanced distribution for all the neurons in the **BP** model.

In **FF**, we find that the bias distribution is strongly imbalanced towards inhibition with a *mean ± std.dev.* value of $-1.66 \pm 1.534$. On the contrary, **BP/FF** and and **BP** show a substantial balance ($-0.017 \pm 0.015$ and $0.003 \pm 0.018$, respectively). Therefore, although the weights of the **FF** model appear imbalanced towards excitation, the average bias $\beta$ is very large and negative, and this might explain why, for this model, we observe such high sparsity values. Conversely, the **BP/FF** model has substantially zero bias (within very small fluctuations), therefore the inhibitory mechanism at work here does not rely upon a negative bias, but on the weights' configuration. From these results, we conclude that not only the training objective (*i.e.,* goodness-based vs. categorical cross-entropy minimisation), but the specific training protocols (**FF** vs **BP/FF**) are determinant in shaping a different interplay between excitation and inhibition, even when similar sparsity levels are observed.

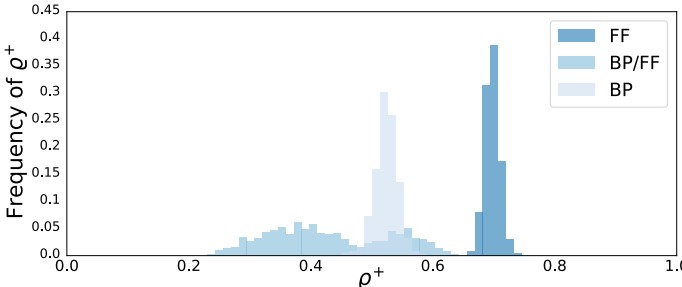

Figure 5: Distribution of $\varrho_i^+$ in Layer 2 (MNIST dataset). In **FF**, the distribution is imbalanced, with most of the population of neurons having $\approx 65 - 75\%$ of excitatory weights. In **BP/FF** the distribution is bimodal with two populations of neurons: one inmbalanced towards excitation (right mode) and the other towards inhibition (left mode). The **BP** model is almost perfectly balanced between excitation and inhibition.

We report the summary statistics $\varrho^+, \beta$ for all the combinations of models, layers and datasets in Table 4 and Table 5. In all settings, we observe a similar picture, with the inhibition mechanisms dominated by negative biases in **FF** and negative weights in **BP/FF**.

In summary, these findings illustrate that although **FF** and **BP/FF** learn similar representations, they achieve sparsity by relying on fundamentally different neuron inhibition strategies.

Table 4: Average fraction of positive weights ($\varrho^+$) for each combination of models, datasets and layers. Results are expressed as *mean $\pm$ std. dev.* over a single training run.

| Dataset | Layer | **FF** | **BP/FF** | **BP** |
|---|---|---|---|---|
| Mnist | 1 | $0.661 \pm 0.031$ | $0.534 \pm 0.028$ | $0.486 \pm 0.022$ |
| | 2 | $0.688 \pm 0.014$ | $0.445 \pm 0.099$ | $0.523 \pm 0.019$ |
| | 3 | $0.882 \pm 0.078$ | $0.535 \pm 0.071$ | $0.52 \pm 0.021$ |
| FashionMnist | 1 | $0.457 \pm 0.099$ | $0.509 \pm 0.018$ | $0.491 \pm 0.021$ |
| | 2 | $0.602 \pm 0.074$ | $0.423 \pm 0.065$ | $0.52 \pm 0.02$ |
| | 3 | $0.416 \pm 0.191$ | $0.433 \pm 0.019$ | $0.52 \pm 0.02$ |
| Svhn | 1 | $0.487 \pm 0.062$ | $0.499 \pm 0.009$ | $0.499 \pm 0.006$ |
| | 2 | $0.521 \pm 0.033$ | $0.427 \pm 0.042$ | $0.504 \pm 0.011$ |
| | 3 | $0.583 \pm 0.096$ | $0.422 \pm 0.031$ | $0.522 \pm 0.023$ |
| Cifar10 | 1 | $0.493 \pm 0.027$ | $0.502 \pm 0.011$ | $0.501 \pm 0.009$ |
| | 2 | $0.489 \pm 0.044$ | $0.43 \pm 0.038$ | $0.513 \pm 0.012$ |
| | 3 | $0.612 \pm 0.046$ | $0.408 \pm 0.034$ | $0.523 \pm 0.02$ |

Table 5: Average bias ($\beta$) for each combination of models, datasets and layers. Results expressed as *mean $\pm$ std. dev.* over a single training run.

| Dataset | Layer | **FF** | **BP/FF** | **BP** |
|---|---|---|---|---|
| Mnist | 1 | $-1.57 \pm 2.029$ | $-0.007 \pm 0.019$ | $0.001 \pm 0.021$ |
| | 2 | $-1.66 \pm 1.534$ | $-0.017 \pm 0.015$ | $0.003 \pm 0.018$ |
| | 3 | $-0.313 \pm 0.151$ | $-0.071 \pm 0.02$ | $0.002 \pm 0.018$ |
| FashionMnist | 1 | $-0.694 \pm 0.706$ | $-0.007 \pm 0.018$ | $0.003 \pm 0.021$ |
| | 2 | $-1.858 \pm 0.489$ | $-0.011 \pm 0.018$ | $0.003 \pm 0.019$ |
| | 3 | $-1.437 \pm 1.213$ | $-0.028 \pm 0.009$ | $0.003 \pm 0.019$ |
| Svhn | 1 | $-0.662 \pm 0.219$ | $-0.033 \pm 0.012$ | $-0.004 \pm 0.013$ |
| | 2 | $-0.959 \pm 0.089$ | $-0.005 \pm 0.011$ | $0.003 \pm 0.012$ |
| | 3 | $-0.962 \pm 0.271$ | $-0.001 \pm 0.016$ | $0.003 \pm 0.011$ |
| Cifar10 | 1 | $-0.402 \pm 0.1$ | $-0.022 \pm 0.008$ | $-0.001 \pm 0.016$ |
| | 2 | $-0.703 \pm 0.132$ | $-0.006 \pm 0.01$ | $0.003 \pm 0.016$ |
| | 3 | $-0.873 \pm 0.076$ | $-0.004 \pm 0.013$ | $0.002 \pm 0.013$ |

## 5   Discussion and conclusions

In many brain circuits, only a small fraction of neurons is active under specific sensory or behavioural conditions. It well established, indeed, that both sensory cortex and the hippocampus exhibit markedly sparse activity. The exact percentages can vary with species, types of stimuli, brain state, measurement technique, and brain area. For example, in rodent primary visual cortex, only about $10-20\%$ of excitatory neurons respond significantly to visual stimuli of varied complexity, from simple ones, such as oriented gratings to complex ones, like movies (Bonin et al., 2011; Miller et al., 2014). In the auditory cortex, the fraction of neurons activated by a particular sound can be as low as $5-15\%$ (Hromádka et al., 2008; Barth & Poulet, 2012). Lastly, hippocampal recordings during animal exploration show that only $20-40\%$ of neurons become active in different environments (Leutgeb et al., 2004). This sparse activity is often organised into *ensembles*, small groups of neurons that consistently co-activate in response to sensory stimuli or during spontaneous activity, and they have been proposed as functional building blocks of sensory processing, memory, and behaviour (Miller et al., 2014; Hebb, 2005; Harris, 2005; György, 2010; Hopfield, 1982; Carrillo-

Reid et al., 2019; Carrillo-Reid & Yuste, 2020; Yuste et al., 2024). We discussed the notion of ensemble in subsection 2.2.

The main finding of our work is that artificial neural networks trained with the Forward-Forward algorithm can elicit sparse representations that share intriguing analogies with the neuronal ensembles found in real brains. We started our investigation by collecting and analysing representations from **FF** networks trained on MNIST, FASHIONMNIST, SVHN, and CIFAR10 and defined, separately for each category, subsets of units (artificial ensembles) that prominently and consistently activated in response to data in such category (subsection 4.2). These category-specific ensembles turn out to be composed of a few active units, which is consistent with the aforementioned experimental findings on the sensory cortex and the hippocampus. Furthermore, when image categories are characterised by a certain degree of visual similarity, the corresponding ensembles often share one or more units (Figure 3, Panel **B**). The fact that single units can appear in multiple ensembles for different categories parallels the idea of mixed selectivity neurons. Mixed selectivity refers to neurons that respond in complex ways to combinations of characteristics or stimuli, thus increasing the dimensionality of population activity and allowing for flexible behaviour (Rigotti et al., 2013; Fusi et al., 2016). Neurons that participate in more than one ensemble can be conceptually viewed as exhibiting mixed selectivity, since they contribute to multiple learned representations simultaneously.

We then tested the ability of trained **FF** models to cope with new data, and observed that activations in response to an unseen input category form, in many cases, a new ensemble with sparsity characteristics similar to those formed for other classes during training (Figure 4, Panels **A** and **B**). We also noticed that the ensembles of unseen categories often show a high level of similarity and integration with the ensembles of the categories of data encountered during training, realised through the sharing of units (see Figure 4, Panel **C** and also the results in Appendix G). Beyond this qualitative similarity with the ensembles formed in response to seen categories, we showed, by training linear probes on representations, that the information content in these activation patterns is almost as high as that of seen categories Table 7. While the **BP** model typically achieves higher decoding performance in this task, it does so by relying on a dense coding scheme. These findings suggest that the ensembles generated by **FF** in response to new data can support zero-shot classification tasks, which is particularly relevant in view of the importance of zero/few-shot learning in human and animal cognitive performance (Lake et al., 2015).

While absent in the **BP** model, the existence of ensembles composed of a few units is not unique to **FF**. It was indeed observed also in **BP/FF** (subsection 4.2), where the ensembles turned out to be of comparable size. This similarity in how representations are organised in **FF** and **BP/FF** is also backed by quantitative analysis with established representation similarity metrics Appendix J. However, despite their similarity at representation level, the **FF** and **BP/FF** models are profoundly diverse, as demonstrated by the different interplay between inhibition and excitation in these models (see subsection 4.5). We observed in this regard that the excitatory/inhibitory (E/I) balance play a key role in the stability of cortical networks and in brain dynamics (Gerstner & Kistler, 2002; Deco et al., 2014).

The sparsity of representations has computational benefits in sensory processing. Olshausen & Field (2004) emphasised that sparsity may be the optimal encoding strategy for neural networks because it is energy efficient. This is especially important for biological neural networks, which operate under metabolic constraints. Sparsity also increases the memory-storage capacity and eases readout at subsequent processing layers. Babadi & Sompolinsky (2014) showed that sparse and expansive coding (*i.e.,* from a lower dimensional sensory input space to a higher dimensional neural representation) reduced the intra-stimulus variability, maximised the inter-stimulus variability, and allowed optimal and efficient readout of downstream neurons. This is the reason why sparse and expansive transformations are widespread in biology, *e.g.*, in rodents (Mombaerts et al., 1996) or flies (Turner et al., 2008).

**Limitations** The limitations of the present work could be addressed by applying similar analyses to more complex datasets and a variety of tasks. To scale to a more challenging dataset may require the replacement of a fully connected network with more suitable architectures trainable with the Forward-Forward protocol, *e.g.,* the CNNs recently introduced in Papachristodoulou et al. (2024) and Sun et al. (2025). This approach could provide insights into how data characteristics influence sparsity levels and the resulting ensemble-

like structures. Moreover, as highlighted in Yang (2023), the choice of hyperparameters potentially affects representation sparsity, as well as the goodness function of choice.

**Future work** Many questions are left open and will be addressed in future works. A closer inspection of the activation patterns within each category will be necessary to test for the co-existence of multiple patterns, with one dominant and possibly many subdominant patterns. We have not yet investigated this *microstructure* (Galán, 2008), leaving it to possible extensions of this work. Based on experimental results indicating the presence of small category-specific ensembles, a promising avenue for future research in this field encompasses exploring model compression through pruning (Blalock et al., 2020), with the design of new strategies based on the relevance of the ensembles, as well as investigating the dynamic evolution of ensemble size and organisation throughout the training process. An especially intriguing perspective, inspired by recent work on optimal sparsity in hippocampal memory models (Shah et al., 2025), suggests that sparsity levels are *dynamic* variables, rather than fixed properties of the network, that can be tuned to the compressibility of sensory inputs to reach optimal performance. Such inquiries hold the potential to shed light on the formation, evolution, interactions, and persistence or replacement of ensembles in artificial neural networks. Lastly, from the analysis of representations learned by the Forward-Forward algorithm, our work suggests novel directions for comparing artificial and biological representations (Barrett et al., 2019; Schrimpf et al., 2018) – particularly for biologically plausible learning algorithms – by leveraging a well-established concept in Neuroscience: that of neuronal ensembles.

## Acknowledgements

We thank Alex Rodriguez for the idea of testing the models on categories unseen during training. We also acknowledge the AREA Science Park supercomputing platform ORFEO and thank the staff of the Laboratory of Data Engineering for technical support.

AA, AC and LB were supported by the project "Supporto alla diagnosi di malattie rare tramite l'intelligenza artificiale" CUP: F53C22001770002 and "Valutazione automatica delle immagini diagnostiche tramite l'intelligenza artificiale", CUP: F53C22001780002. AA and AC were supported by the European Union – NextGenerationEU within the project PNRR "PRP@CERIC" IR0000028 - Mission 4 Component 2 Investment 3.1 Action 3.1.1.

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

## A   Computational resources

Training and subsequent experiments were conducted on an NVIDIA DGX A100 system. The system is equipped with 8 NVIDIA A100 GPUs, interconnected by NVLink technology, two AMD EPYC 7742 64-core CPUs, 1TB of RAM, and a 3TB NVME storage configured in RAID-0. Each GPU is equipped with 6912 CUDA cores, 432 Tensor cores and 40 GB of high-bandwidth memory.

## B   Data

The MNIST dataset consists of pictures of handwritten Arabic numerals, from 0 to 9, each represented as a grayscale image of size $28 \times 28$. FASHIONMNIST has been designed as a drop-in replacement to MNIST, offering a more challenging classification task. It consists of ten classes of clothing items, still represented as grayscale images with a resolution of $28 \times 28$. Both datasets provide 60000 training and 10000 test images, balanced in terms of per-class numerosity.

SVHN contains coloured images of digits from house numbers, captured by Google StreetView. The images are composed of $32 \times 32$ RGB-encoded pixels. This dataset is slightly larger than the previous two, as it contains 73257 data-points in the training set and 26032 in the test set.

The SVHN images have been cropped in order to center the digit of interest within the frame. However, the presence of adjacent digits and other distracting elements, that have been kept within the images, introduces an additional layer of complexity when compared to MNIST and FASHIONMNIST, where the subjects are prominently displayed against a uniform black background. The CIFAR10 consists of 60000 coloured natural images categorised in 10 balanced classes. The dataset is split in 50000 training images and 10000 test images. Each image, like SVHN has a resolution of $32 \times 32$ for each channel. Compared to previous datasets, this is the most challenging one for a fully connected network. The dataset split employed is provided by the TORCHVISION framework (TorchVision maintainers and contributors, 2016).

## C   Training details

All our models (**FF**, **BP/FF** and **BP**), on all datasets (MNIST, FASHIONMNIST, SVHN and CIFAR10), have been optimised using Adam (Kingma & Ba, 2017) with $\beta_1 = 0.9$ and $\beta_2 = 0.999$, implemented in PYTORCH (Paszke et al., 2019). A hyperparameter search has been performed to achieve sufficient accuracy for each model across all datasets. Every model was trained using batches of size 1024.

Table 6: Hyperparameters selected to train our models.

| Model | | MNIST | FASHIONMNIST | SVHN | CIFAR10 |
|---|---|---|---|---|---|
| **FF** | Epochs | 1200 | 100 | 1000 | 1000 |
| | Learning rate | 0.01 | 0.01 | 0.0001 | 0.0001 |
| **BP/FF** | Epochs | 300 | 300 | 200 | 200 |
| | Learning rate | 0.0001 | 0.0001 | 0.0001 | 0.0001 |
| **BP** | Epochs | 80 | 80 | 80 | 80 |
| | Learning rate | 0.0001 | 0.0001 | 0.0001 | 0.0001 |

## D   Statistical test of ensemble assignment

To assign a unit $i$ to a class-ensemble $c$, we compare its average activation on correct responses $\overline{x_{i,c}}$ against its leave-one-out average $\mathrm{LOO}_{i,c}$ (as described in subsection 3.5). We define

$$\delta_{i,c} = \overline{x_{i,c}} - 2 \cdot \mathrm{LOO}_{i,c}.$$

If $\delta_{i,c} > 0$, the unit is assigned to the ensemble for class $c$. To assess the statistical significance of each assignment, we compute a $p$-value by building an empirical null distribution. Specifically, we shuffle each

relevant representation matrix row-wise 200 times, recalculate $\delta_{i,c}^{\mathrm{rand}}$ for each shuffle, and then define the $p$-value as the fraction of these shuffled values exceeding the observed $\delta_{i,c}$. We applied this test to every unit assigned to an ensemble across all model/dataset/layer/run combinations summarised in Table 3, totaling $\approx 457,000$ units. Of these, only 500 (about 1 in 1000) had $p > 0.05$. Most of those higher-$p$ units occurred in Layer 2 of **BP/FF** on CIFAR10, representing $\approx 2.6\%$ of the assigned units in that specific configuration.

## E    Activation patterns in deeper layers

In subsection 4.2 we claimed that in **FF** and **BP/FF** the images of a given category activate consistently a small set of units that we named ensembles, that share similarities to what is observed in sensory cortices. We reported in Figure 1 the activation map for Layer 1 (the first hidden layer) of **FF** trained on the MNIST dataset, and observed that very sparse ensembles emerge. In this section we show, in a similar fashion, the representations for Layers 2 and 3 (Figure 6 and Figure 7, respectively). We find high sparsity also for deeper layers of this specific network; a qualitatively similar conclusion is reached for **FF** models trained on FASHIONMNIST, SVHN and CIFAR10. In **BP/FF** models a similar sparsity levels are observed, with the exception of the last layer that turns out to be non-sparse in all the datasets considered (see Table 3).

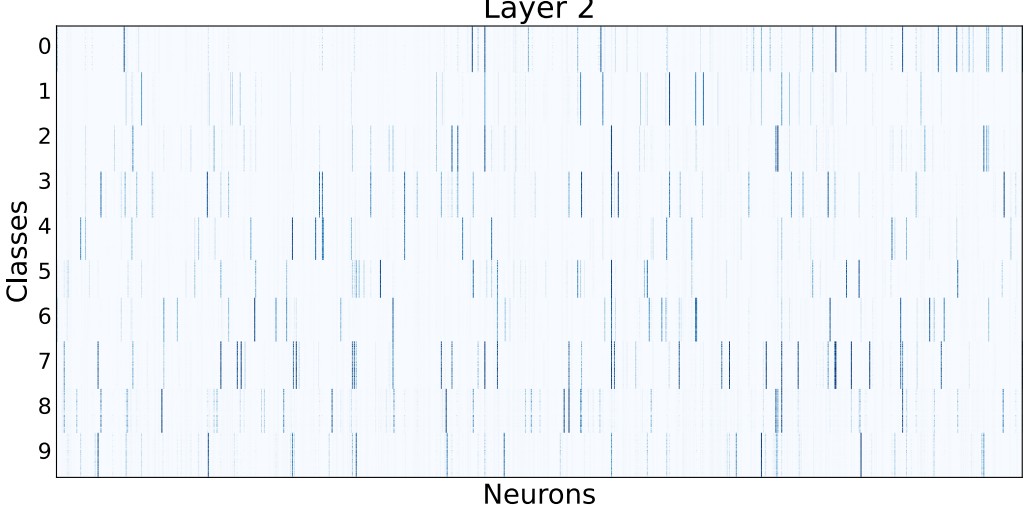

Figure 6: Activation patterns in a Multi-Layer Perceptron trained with the Forward-Forward algorithm, on the MNIST dataset. The image represents the activation map for neurons in Layer 2 for all images, grouped by class. A blue dot in position $(x, y)$ indicates that neuron $x$ is activated by input $y$; colour scale represents the intensity of such activation (incorrectly classified samples have been removed). Horizontal bands mark different categories; dark blue vertical lines mark active neurons. Each input category activates consistently a specific sets of neurons (ensemble). The sparsity measured according with the definition provided in subsection 3.5 is 0.84.

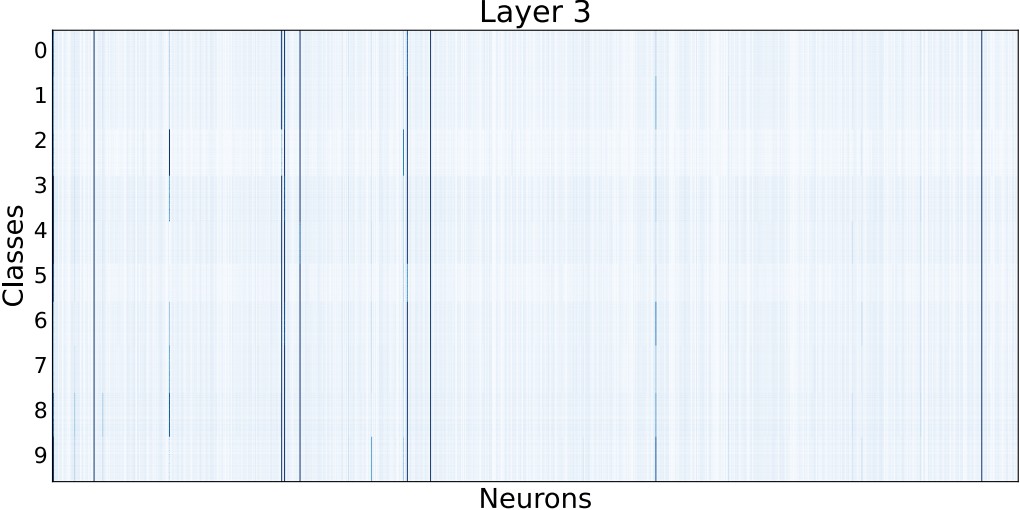

Figure 7: Activation reported as in Figure 6, for Layer 3. Notice that there are only few units that activates significantly and does not play a role in discriminating categories. The role of this layer, in this experiment seems, not related to the classification task. Despite the low number of active units, the sparsity level of the representation is lower than that of Layer 2 ($S = 0.67$), due to the noise of the inactive units.

## F  Activation patterns in different models

In Figure 1 (Panel **C**) we show the activation patterns in Layer 1 of **FF** trained on MNIST. For the purpose of a qualitative comparison, we show here analogous patterns for **BP/FF** and **BP** (see Figure 8 and Figure 9).

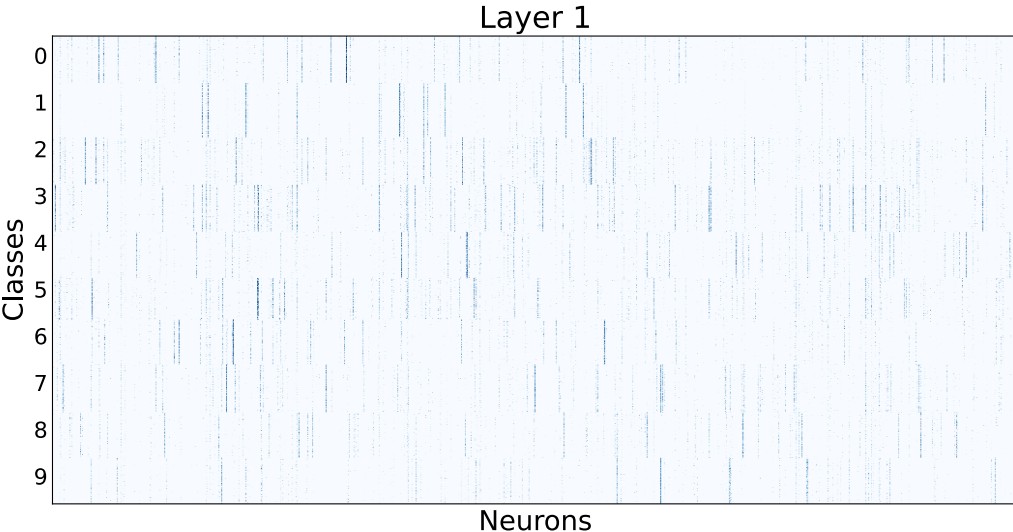

Figure 8: Activation pattern in Layer 1 of the **BP/FF** model trained on the MNIST dataset. The sparsity measure is 0.89, comparable with the correspondent first layer of the **FF** model, reported in Figure 1, Panel **C**.

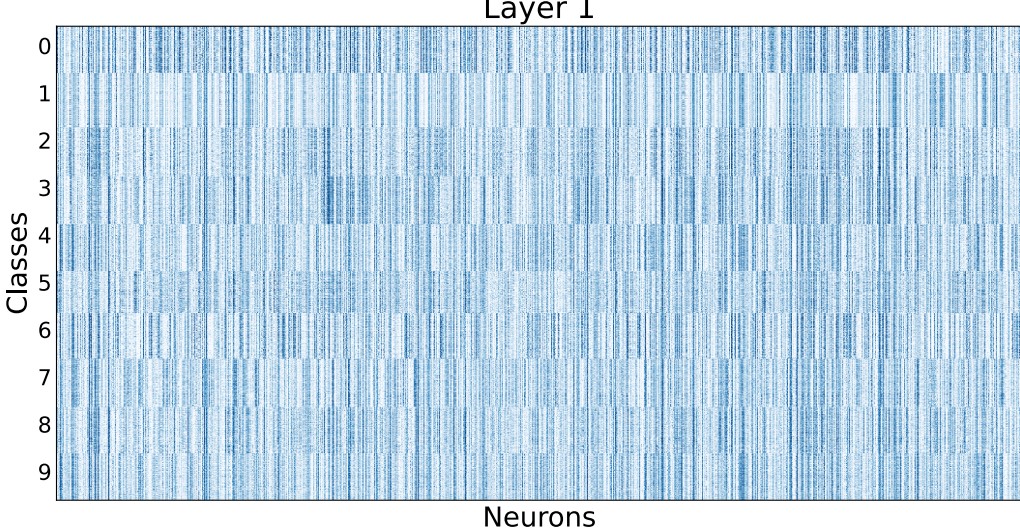

Figure 9: Activation pattern in Layer 1 of the **BP** model trained on the MNIST dataset. The sparsity measure is 0.32 (non-sparse representation), about $\frac{1}{3}$ of the sparsity level measured in the analogous experiment with **FF** and **BP/FF**.

## G   Further results on representations of unseen categories and their ensembles

We showed in subsection 4.4 that a **FF** model trained on the FASHIONMNIST dataset – deprived of one category – can respond at test time to this unseen category with an ensemble (Figure 4).

We report here the results of similar experiments, removing one category at a time. It turns out that, in each of the ten possible cases (we performed a single run for each category), the representations of the unseen category form an ensemble; we show three examples in Figure 10, different from the example shown in (Figure 4). It is with this situation in mind that we refer to "the ensembles related to unseen categories".

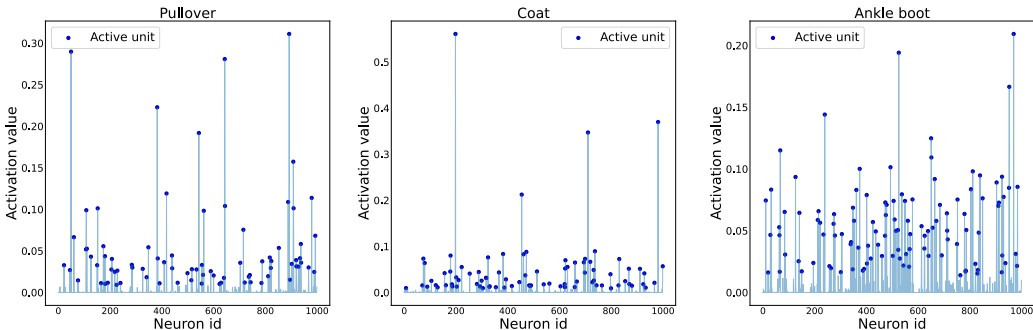

Figure 10: Ensembles elicited by the **FF** model trained on FASHIONMNIST deprived of one category (we show three examples: `Pullover`, `Coat` and `Ankle boot`). We report for the three categories, the activation value of each neuron in the first hidden layer (Layer 1), averaged on all images of the unseen category. Neuron index on the $x$ axis; average activation on the $y$ axis. Blue dots indicate units that are considered active according to the method described in subsection 3.5.

When an unseen category forms an ensemble, it generally exhibits a high level of integration with the ensembles associated with the categories encountered during training. This integration implies that it can share common units with ensembles belonging to related categories. We show in Figure 11 how the ensembles of missing categories (same examples as in Figure 10) integrate – by sharing units – with the other ensembles.

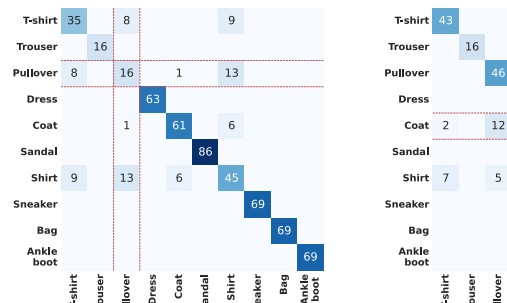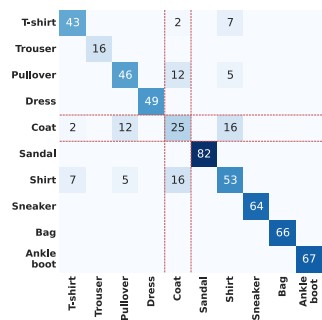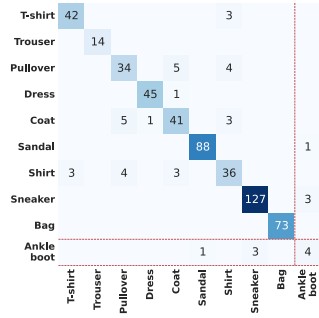

Figure 11: Shared units between the ensembles of unseen categories and the ensembles of categories seen during training (stripes delimited by the red lines). The results for `Pullover`, `Coat` and `Ankle boot` are shown.

Overall, these result relates to biological neural networks (Yoshida & Ohki, 2020; Yuste, 2015), where ensembles appear to be the functional building block of brain representations even in the absence of known stimuli.

## H Performance on unseen categories with linear probes

In this section, we explore the ability of a linear probe to discriminate between categories, including those unseen during training, leveraging the models' internal representations. Focusing on the FASHIONMNIST dataset, we trained 10 models – each excluding a different category – for all of **FF**, **BP/FF**, and **BP**. A linear classifier was then trained on the training set representations. We note here that for this experiment **FF** and **BP/FF** were provided with the same data as **BP**, *i.e.,* images without any pixels encoding positive and negative labels. Our findings, reported in Table 7 reveal that the linear probes successfully recover good accuracy levels in almost all cases. This result holds consistently across all categories and models. We also report in Table 8 the performance of linear probes averaged across all categories, seen and unseen, including a comparison with baseline models. This shows that the decoding performance of linear probes trained on models in which a category is held out during training is close to the original one (without any held out category).

Table 7: Linear probe accuracy on the missing category for models trained without that category.

| Model | Missing category | | | | | | | | | |
|---|---|---|---|---|---|---|---|---|---|---|
| | 0 | 1 | 2 | 3 | 4 | 5 | 6 | 7 | 8 | 9 |
| **FF** | 0.803 | 0.901 | 0.719 | 0.817 | 0.714 | 0.869 | 0.508 | 0.876 | 0.920 | 0.889 |
| **BP/FF** | 0.787 | 0.939 | 0.699 | 0.856 | 0.742 | 0.894 | 0.454 | 0.836 | 0.947 | 0.937 |
| **BP** | 0.863 | 0.962 | 0.721 | 0.933 | 0.843 | 0.967 | 0.632 | 0.945 | 0.970 | 0.957 |

Table 8: Linear probe accuracy - averaged across all the categories - for models trained without one category. The Avg column reports the average across all the ten cases. The Baseline corresponds to the accuracy achieved by the model when all categories are included during training.

| Model | Missing category | | | | | | | | | | Avg | Baseline |
|---|---|---|---|---|---|---|---|---|---|---|---|---|
| | 0 | 1 | 2 | 3 | 4 | 5 | 6 | 7 | 8 | 9 | | |
| **FF** | 0.854 | 0.857 | 0.847 | 0.854 | 0.849 | 0.851 | 0.843 | 0.851 | 0.858 | 0.862 | 0.852 | 0.849 |
| **BP/FF** | 0.868 | 0.869 | 0.859 | 0.864 | 0.863 | 0.866 | 0.859 | 0.856 | 0.866 | 0.858 | 0.864 | 0.877 |
| **BP** | 0.880 | 0.890 | 0.883 | 0.885 | 0.883 | 0.888 | 0.887 | 0.887 | 0.886 | 0.889 | 0.886 | 0.892 |

## I Enforcing sparsity in the BP model

In the **FF** and **BP/FF** models sparsity emerges without any explicit regularisation or constraint. Instead, if one wanted to promote sparse representations in the **BP** model, one established way is by means of $\ell_1$ regularisation on the activations (Georgiadis, 2019). In this section, we present the results obtained by training a **BP** model using the same hyperparameters as those employed in the main analysis and $\ell_1$ regularisation on the activations, with weight set to $2.5 \times 10^{-6}$. We measure the layer-wise sparsity in the MNIST and FASHIONMNIST datasets, and observe (Table 9) sparsity values comparable – and often higher – than the ones that spontaneously emerge with **FF** and **BP/FF**, reported in Table 2.

Table 9: Sparsity of the **BP** model with $\ell_1$ norm regularisation applied to layer activations.

| Model | Layer | MNIST | FASHIONMNIST |
|---|---|---|---|
| **BP regularised** | 1 | 0.971 | 0.955 |
| | 2 | 0.802 | 0.787 |
| | 3 | 0.813 | 0.781 |

However, despite having very high sparsity levels, the representations learned by **BP** with $\ell_1$ regularisation and by unregularised **FF** display significant differences. Figure 12 reports the Jaccard similarity between MNIST class ensembles (computed using the procedure of subsection 3.5) for the first layer of the regularised **BP** model (left) and the **FF** model (right). **FF** ensembles are highly specific, as they are completely disjoint, while in the case of regularised **BP** there is a non-zero overlap for any pair of classes, regardless of their visual dissimilarity.

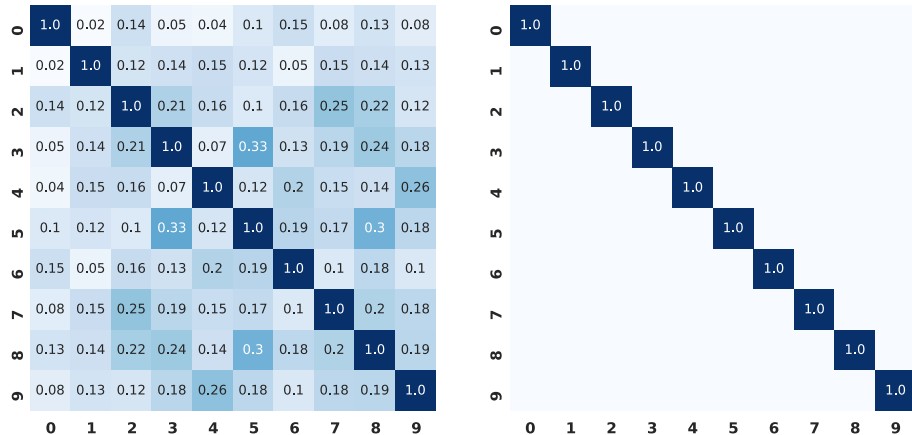

Figure 12: Jaccard similarity between first-layer ensembles on the MNIST dataset. Left: **BP** model with $\ell_1$ regularisation. Right: **FF** model.

## J    Representation similarity

### J.1    Model comparison

In this section, we analyze the similarity between representations produced by **FF**, **BP/FF** and **BP**. We employ three established representation similarity metrics, namely SVCCA (Raghu et al., 2017), CKA (Kornblith et al., 2019) and Distance Correlation (dCor) (Székely et al., 2007). We consider 5 training runs with independent weight initialisations for each model and compare representations layer-by-layer. The results, reported in Table 10, show that:

- CKA and dCor exhibit greater variability than SVCCA across models and layers. We hypothesize that this may be because SVCCA is a linear metric, making it less informative in the presence of nonlinear correlation patterns;

- The first layer is almost always the most similar, possibly due to the fact that it is the one closest to the input, which is shared between models;

- Focusing on the second layer, **FF** and **BP/FF** are consistently the most similar pair of models according to CKA and dCor. The same does not apply for the third layer, which is in fact non-sparse in **BP/FF** (Table 3).

Table 10: Representation similarity between models. Results are averaged over 5 runs with independent random weight initialisation for each configuration.

| Dataset | Metric | FF v BP/FF | | | FF v BP | | | BP/FF v BP | | |
|---|---|---|---|---|---|---|---|---|---|---|
| | | 1 | 2 | 3 | 1 | 2 | 3 | 1 | 2 | 3 |
| Mnist | SVCCA | 0.48 | 0.38 | 0.45 | 0.55 | 0.47 | 0.52 | 0.48 | 0.39 | 0.45 |
| | CKA | 0.79 | 0.53 | 0.04 | 0.53 | 0.49 | 0.23 | 0.62 | 0.43 | 0.02 |
| | dCor | 0.85 | 0.90 | 0.13 | 0.61 | 0.68 | 0.45 | 0.70 | 0.64 | 0.17 |
| FashionMnist | SVCCA | 0.55 | 0.35 | 0.33 | 0.53 | 0.36 | 0.37 | 0.47 | 0.37 | 0.41 |
| | CKA | 0.60 | 0.63 | 0.24 | 0.51 | 0.40 | 0.34 | 0.60 | 0.40 | 0.12 |
| | dCor | 0.80 | 0.76 | 0.10 | 0.59 | 0.51 | 0.50 | 0.81 | 0.59 | 0.11 |
| Svhn | SVCCA | 0.57 | 0.55 | 0.55 | 0.56 | 0.53 | 0.49 | 0.58 | 0.50 | 0.51 |
| | CKA | 0.47 | 0.40 | 0.27 | 0.32 | 0.17 | 0.04 | 0.72 | 0.21 | 0.01 |
| | dCor | 0.60 | 0.46 | 0.09 | 0.45 | 0.27 | 0.13 | 0.90 | 0.38 | 0.03 |
| Cifar10 | SVCCA | 0.54 | 0.53 | 0.54 | 0.53 | 0.50 | 0.50 | 0.56 | 0.46 | 0.47 |
| | CKA | 0.46 | 0.46 | 0.17 | 0.37 | 0.17 | 0.04 | 0.65 | 0.25 | 0.02 |
| | dCor | 0.64 | 0.53 | 0.06 | 0.49 | 0.22 | 0.07 | 0.86 | 0.39 | 0.06 |

### J.2    Layer comparison

In Figure 13 we report the CKA similarity between different layers of **FF** models, averaged over 10 independent training runs. Across all datasets, similarity is noticeably higher for layers 1 and 2 than for layers 2 and 3. The only partial exception is FashionMnist, that displays a much narrower gap. These results align with those in Table 2: FashionMnist is the only dataset in which sparsity in the second layer is lower than in the third one.

## K    Forward-Forward with $\ell_1$ goodness function

We investigated the effect of a different choice of goodness function by switching to the $\ell_1$ norm. Consequently, we adjusted the normalisation for subsequent layers, performed according to the $\ell_1$ norm as well.

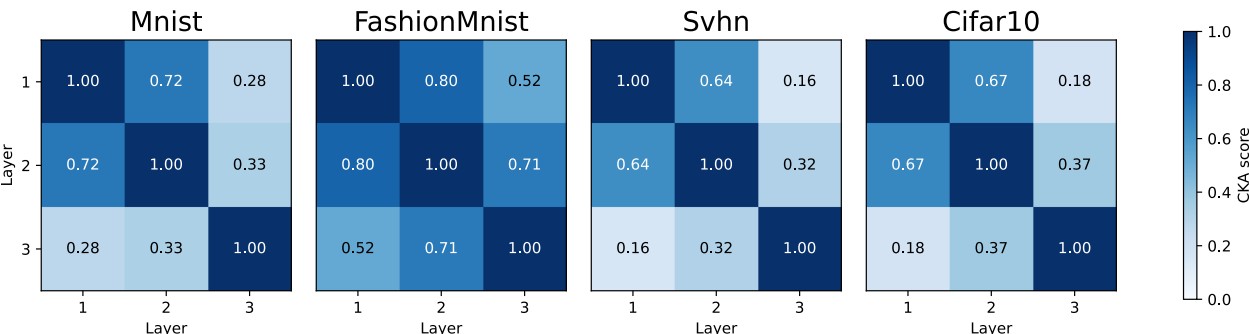

Figure 13: CKA similarity between layers in the **FF** model, averaged over 10 independent training runs.

We trained 10 instances of **FF** and of **BP/FF** on MNIST and on FASHIONMNIST datasets. The hyperparameters were set as follows: learning rate: 0.001, epochs: 300, batch size: 1024. With this setup, we train the **FF** and **BP/FF** models on the MNIST and FASHIONMNIST datasets.

Our observations indicate that the accuracy achieved by both the **FF** and **BP/FF** models is comparable with the $\ell_2$ results reported in Table 1. The level of sparsity is consistent with the findings presented in Table 2. However, in the case of the **BP/FF** model, the third layer exhibits significantly higher sparsity compared to when using the $\ell_2$ norm. Despite this increase, sparsity remains insufficient for recognizing robust and consistent ensembles. The average fraction of active units per layer is reported in Table 13. Results regarding ensemble overlap between visually similar classes in the **FF** model are reported in Figure 14. The findings of subsection 4.3 remain valid when employing the $\ell_1$ norm as a goodness function.

Table 11: Test-set classification accuracy for the models **FF** and **BP/FF**, using a goodness function based on the $\ell_1$ norm. Results expressed as *mean $\pm$ std. dev.* over 10 runs with independent randomised weight initialisation.

| Dataset | **FF** | **BP/FF** |
|---|---|---|
| MNIST | $0.949 \pm 0.002$ | $0.965 \pm 0.001$ |
| FASHIONMNIST | $0.859 \pm 0.002$ | $0.865 \pm 0.002$ |

Table 12: Average sparsity for **FF** and **BP/FF** with $\ell_1$ goodness function, computed according to the definition given in subsection 3.5. Results are expressed as *mean $\pm$ std. dev.* computed over 10 runs with independent random weights initialisation.

| Model | Layer | MNIST | FASHIONMNIST |
|---|---|---|---|
| **FF** | 1 | $0.887 \pm 0.002$ | $0.83 \pm 0.001$ |
| | 2 | $0.61 \pm 0.005$ | $0.647 \pm 0.007$ |
| | 3 | $0.61 \pm 0.012$ | $0.532 \pm 0.012$ |
| **BP/FF** | 1 | $0.944 \pm 0.002$ | $0.921 \pm 0.003$ |
| | 2 | $0.915 \pm 0.005$ | $0.919 \pm 0.003$ |
| | 3 | $0.441 \pm 0.009$ | $0.408 \pm 0.011$ |

Table 13: Average fraction of units taking part in ensembles for **FF** and **BP/FF** with $\ell_1$ goodness function. Ensemble sizes are averaged across all categories, divided by the number of neurons in a layer, and then expressed in %. Ensembles are defined according to the LOO method presented in subsection 3.5. Results are expressed as *mean $\pm$ std. dev.*. In the third layer of **BP/FF** the representation is non-sparse.

| Model | Layer | MNIST | FASHIONMNIST |
|---|---|---|---|
| **FF** | 1 | $4.22 \pm 0.09$ | $7.48 \pm 0.11$ |
| | 2 | $19.93 \pm 0.53$ | $19.44 \pm 0.53$ |
| | 3 | $17.24 \pm 0.85$ | $23.09 \pm 1.15$ |
| **BP/FF** | 1 | $3.88 \pm 0.16$ | $5.93 \pm 0.21$ |
| | 2 | $3.36 \pm 0.29$ | $5.26 \pm 0.32$ |
| | 3 | - | - |

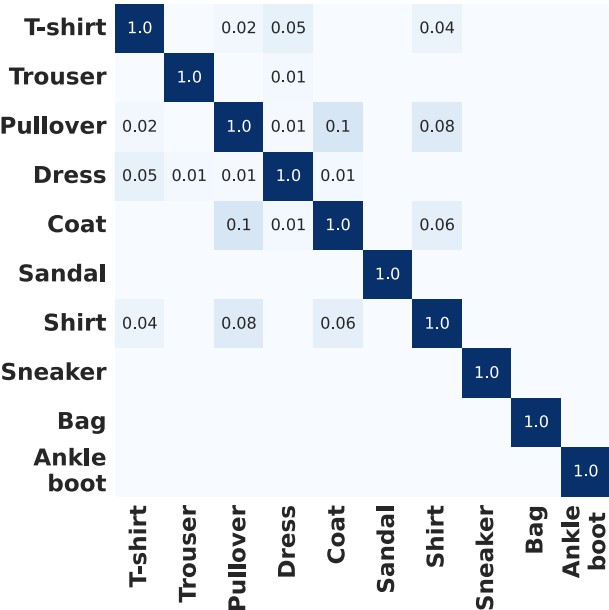

Figure 14: Jaccard similarity index between first-layer ensembles. Results obtained using the $\ell_1$ norm as a goodness function in the **FF** model on the FASHIONMNIST dataset.

