# OpenReview forum: "Emergent representations in networks trained with the Forward-Forward algorithm"
_TMLR — Accepted by TMLR_

### Review · Reviewer_DE1Q · 2024-12-18

**Summary Of Contributions:**

The paper investigates the internal representations formed by neural networks when trained using the Forward-Forward algorithm. The authors find out that these internal representations are highly sparse and organized into category specific ensembles mirroring the behavior of neurons in biological brains. They also find that semantically similar classes activate overlapping ensembles with shared neurons.

They run a set of experiments on three variants of model setups

- Forward forward algorithm (FF)
- Backprop trained using forward-forward objective (BP/FF)
- Standard backprop with cross entropy loss (BP)

They show that FF and BP/FF exhibit high sparsity and form ensembles, while BP does not.

The main conclusion of this paper is that neural networks trained using Forward-Forward algorithm offers a more biologically plausible learning mechanism than standard backprop algorithm.

**Audience:**

Yes

**Claims And Evidence:**

Yes

**Requested Changes:**

I would recommend the authors to at least address some of the points mentioned in the weaknesses section.

**Strengths And Weaknesses:**

**Strengths:**

- Forward-forward algorithm is a relatively new and less studied topic. This paper helps improve our understanding of it.
- The paper is easy to follow and understand

**Weaknesses:**

- The authors only test L2 norm as the goodness function despite mentioning that sparsity seems to be a result of the objective function rather than the FF algorithm itself. There should have been a section in the paper studying the impact of different goodness functions on the three model setups.
- I would also be interested in seeing how standard regularization methods compare to the implicit regularization in FF models. For example, how sparse are the representations of standard backprop with cross entropy if regularization is used? Are neural ensembles formed if we regularize a standard model? Are the same set of neurons becoming active?
- The authors only show results for test set, but it would have been interesting to see if the same levels of sparsity and category specific ensembles are being created across the training data too.  There should be a comparative analysis on training vs test data highlighting the differences and similarities.
- In Fig 3 of the paper, the authors have shown a Jaccard Similarity Matrix highlighting that semantically similar classes have shared neurons. But I feel that the values in the matrix are too low (in the range of 0.07 - 0.16) to have a strong case for being similar. I would have expected a higher similarity index for semantically similar classes like pullover and coat. Is there any explanation to why the values are relatively small?
- While the authors compare the representations of different model setups using excitatory and inhibitory connections, it would be interesting to see how similar the different model representations are using CCA/CKA similarity measures.
- As layers are trained separately in case of FF, it would be insightful to have a more in-depth comparison of the differences and similarities between the three layers of the trained FF model

---

### Review · Reviewer_2pNz · 2024-12-30

**Summary Of Contributions:**

The paper proposes to analyze the sparsity patterns in the activation space learned by the Forward-Forward (FF) algorithm comparing with the standard back propagation (BP) algorithm. The paper proposes to analyze three algorithms the FF algorithm, the BP algorithm optimizing cross entropy loss and the BP algorithm optimizing the goodness function (sum of squared activations).  The sparsity is mearured on networks trained on MNIST, FashionMNIST, SVHN, and CIFAR10  Its contributions can be summarized as follows:

- The papers demonstrates empirical evidence that optimizing for a goodness loss function both locally (FF algorithm) and globally (BP applied on the goodness loss function) induces sparsity (measured as the l1/l2 ratio of the sample activations), implicitly.

- The sparsity can be measured at the class (category) level,  with individual units contributing to multiple ensembles when visual features are shared. unveiling sparse ensembles w.r.t. to class in activation space.

- The FF-trained networks can form new sparse ensembles for unseen categories, explaining them as a sparse combination of in distribution categories. This supports the potential for zero-shot classification and out of distribution generalization applications.

**Audience:**

Yes

**Broader Impact Concerns:**

There are no ethical implications to be addressed according to my judgement.

**Claims And Evidence:**

Yes

**Requested Changes:**

**Clarity**

To improve clarity of the paper, I suggest the following changes:

- List of contributions well specified at the end of the introduction. This helps readers quickly identify the key outcomes of the work and how it contributes to the field.

- A one liner at the end of each experimental section, stating the takeaway of the experiment. This will make the results more accessible.

- A small visualization explaining how an ensemble is computed (while simple) could help in providing a more direct understanding of the section.

**Related work**

- Discuss the relationship between findings and theoretical results in Yang (2023): the paragraph should highlight how your results align with, extend, or contrast the findings in Yang (2023)

**Experiments**

Some of these aspect might be discussed or showed:

- Do the results depend on the goodness function chosen?

- How the sparsity patterns changes across different initializations (seeds)?

- In the ensemble setting, does the sparsity level of each class tell something about the category wise distributions? I.e., e.g. in Figure 2, if one class shows more sparsity than another, what does this tell us? Can it be related with class imbalance?

- broader:
    - It would be  interesting to see how the performance on unseen categories changes between the FF, BP and BP/FF algorithms, by e.g. training a linear probe on the representations.

    - Can the setting of the FF algorithm and the sparsity result  related with networks trained with weight decay and BP?

   -  How the sparsity pattern is linked to overparametrization condition of the network? I.e. in overparametrized settings is there less of specialized neurons shared across samples (less sparsity) as the network tends to memorize more single examples? One way to see this would be to train the same network on subsamples of the dataset.


By providing answers to some of the points above and incorporating the corresponding analyses, the paper would provide a more comprehensive understanding of the findings and their implications.

**Strengths And Weaknesses:**

**Strenghts**

- The paper is well written and simple.

- The paper demonstrates a good empirical analysis of the differences in representation learned between networks trained using FF and BP, showing emergence of sparse ensembles in the former case.

- The emergence of sparsity patterns shows the potential of FF to interpretability and OOD generalization applications.

**Weaknessess**

- Some biological claims are not well enough supported and the connection is loose. E.g. the link with the Neuronal ensembles literature lack deeper discussion nor any theoretical or empirical evidence of brain activations resembling sparse ensembles learned by the FF algorithm.

-  The poor scalability of the FF algorithm to more complex networks and dataset limits the practical impact of future promising direction (e.g. OOD generalization)

---

### Review · Reviewer_3UYx · 2025-01-25

**Summary Of Contributions:**

The authors study the emergent internal representations that evolve using the Forward-Forward algorithm using novel techniques inspired by Neuroscience.

**Audience:**

Yes

**Broader Impact Concerns:**

While no Broader Impact Statement is present, this work doesn't pose any broader ethical concerns.

**Claims And Evidence:**

Yes

**Requested Changes:**

1. Important problem and interesting approach. The paper would be of interest to the ML and Neuroscience Communities but the datasets are too easy for an ML paper and connections to Neuroscience is too limited for a comp neuro paper. I currently recommend a weak accept.
2. Fig 1: Describe in the fig legend that the class label is embedded on the image itself.
3. The neuroscience references are rather limited. Please see several recent works by the Stefano Fusi Lab. See “Bio-inspired neural networks implement different recurrent visual processing strategies than task-trained ones do” by Lindsay for insights on metrics to add.
4. Is ImageNet training possible? if not, please add an estimate of the runtime, or a justification for excluding it.
5. Fig 3: Is it semantic similarity or visual similarity that causes a higher overlap? Fig 3B: the similarity indices are rather small. There are only 4 off diagonal entries and they are all very small. I am not sure the claim that ‘semantic similarity causes a high overlap’ holds. Sandals and sneakers would be expected to have a higher overlap by this logic. There’s no reason semantic similarity should matter here. The network is not trained on any language input, nor any class co-occurrence data. Visual similarity should be the driver of the high similarity index, not semantic similarity. There are studies that try to tease apart the roles of semantic and visual similarity, e.g. Yeh and Peelen, 2022; Yeh, Thorat, and Peelen, 2024. At least, a simple experiment to quantify visual similarity and semantic similarity should be done. Word embedding distances for the categories could be used for semantic similarity, or something like object2vec from Bonner and Epstein, 2021. For image similarity, the LPIPS score could be used. These may not be the best metric anymore, please choose a reasonable metric.
6. Use RSA to compare the representations of the three model types and other previously used methods, such as BrainScore.
7. Carandini and Heeger (C&H) is divisive normalization wrt neighboring neurons. The normalization described in the methods is across all neurons in the layer, which is not the normalization C&H meant (which is division by local neighbors of the neuron). The authors have used C&H to claim biological plausibility, but it is a stretch because what the authors did is not exactly what C&H talk about.
8. Slight nitpicking but embedding the image class on the image itself is not exactly biologically plausible either.
9. Table 1: why is the std dev much higher for BP for SVHN, compared to the other two models?
10. Fig 2: Why does BP/FF layer 3 differ from FF in terms of sparsity, but not the other 2 layers?
11. Just comparing sparsity index is not very convincing. If citing Yoshida and Ohki, please include some of their analyses, like linearly decodable info.
12. Fig 5: Why only layer 2?
13. “The idea is that a neuron should be considered active and part of an ensemble if it activates consistently and selectively when the network receives input data that belongs to that category”. Wouldn’t ‘selectively’ imply that neurons should not be shared across categories, then? Selectivity in other contexts in neuroscience differs than what the authors imply here (e.g. see Rust and DiCarlo, 2010, Madan et al., 2022).
14. Equation numbering would be good (if allowed by the journal).
15. The definition of the category-wise means in the section defining ensembles uses a symbol x_j which is defined separately for each category, so the symbol should also specify the category (something like x_j,c)
16. For defining x_j, why are only the correctly classified images M_c used? One could argue that these neurons are responsible for the network saying that category C is present in an image, but then this computation should include images of categories other than C, where the network falsely detects C.
17. Classifying a neuron as active in an ensemble compares the mean activations with the means of means, but this comparison could involve significance testing with a p-value threshold
18. If classifiers are trained based on responses of category ensembles, is the entire ensemble required to get the maximum decoding accuracy for that category?
19. Fig 4: It makes sense that a network would try to project unseen images to the same latent space that works for most of its seen categories. I would expect backprop-trained models to do so too. Whether Fig 4 tells us something non-trivial can only be assessed by comparing against the other two model types. Additionally, sandal is close enough to categories seen while training, so it makes sense that the network uses a strategy similar to other images. What would the results look like for truly out-of-distribution images, like a spaceship? I am also not sure what the authors mean by a ‘valid’ ensemble.
20. Across figures the choice of dataset keeps fluctuating between MNIST and FashionMNIST

**Strengths And Weaknesses:**

Strengths:
1) Timely problem.
2) Of interest to both ML and Comp Neuro communities
3) Novel neuroscience-inspired methods that can be used to study internal representations in deep neural networks.

Weaknesses (see requested changes for more detail):
1) Connection to neuroscience is intriguing but limited (mainly focusing on sparsity and sharing of neurons across stimuli type).
2) No comparison using several previously established methods for comparing artificial and biological networks
3) Limited to easier datasets. Intuitions may not hold for larger datasets like ImageNet.

---

### Decision · Action_Editor_HC2H · 2025-03-03

**Recommendation:** Accept as is

**Comment:**

This work investigates the representations that emerge through use of the Forward-Forward training algorithm. This work is of interest to computational neuroscientists and those interested in the science of machine learning. The results are convincing and, while focused on small-scale datasets, will open up new research directions. Given that all 3 reviewers recommend acceptance as is, I am recommending the same.

**Audience:**

All 3 reviewers agreed that the work investigated a novel training paradigm that would be of broad interest to TMLR's audience. All 3 reviewers maintained this opinion in their official recommendation.

**Claims And Evidence:**

All 3 reviewers agreed, after their initial reviews, that the submission made claims that are supported by evidence. All 3 reviewers maintained this opinion in their official recommendation.